# Effects of Laser Cutting Parameters on the Magnetic Properties of 50W350 High-Grade Non-Oriented Electrical Steel

**DOI:** 10.3390/ma16041642

**Published:** 2023-02-16

**Authors:** Qian Xiang, Lin Cheng, Kaiming Wu

**Affiliations:** 1The State Key Laboratory of Refractories and Metallurgy, Hubei Province Key Laboratory of Systems Science on Metallurgical Processing, International Research Institute for Steel Technology, Collaborative Center on Advanced Steels, Wuhan University of Science and Technology, Wuhan 430081, China; 2Wuhan Iron and Steel Co., Ltd., Wuhan 430081, China

**Keywords:** laser cutting, high-grade non-oriented electrical steel, magnetic properties, specific total loss

## Abstract

Based on the application demand of laser cutting technology in non-oriented electrical steel, the influencing mechanisms of laser cutting parameters on the magnetic properties of 50W350 high-grade non-oriented electrical steel were investigated in this work. The specific total loss was utilized to evaluate the quality of cutting methods and the cutting parameter combinations. The results showed that the deterioration of the specific total loss was mainly due to the increase in hysteresis loss. Compared to traditional mechanic shearing, laser cutting generally degrades the magnetic properties under the evaluation index Δ*P_1.0/50_*. However, in some cases, laser cutting is superior to the mechanic shearing method under the evaluation index Δ*P_1.5/50_*. The main parameters related to laser cutting exhibited complicated influencing mechanisms on the specific total loss of 50W350 high-grade non-oriented electrical steel. However, based on the results of the experiments designed using the Box–Behnken model, the laser cutting parameters were optimized and the evaluation indexes have been significantly improved.

## 1. Introduction

High-grade non-oriented silicon steel used in generators, motors, electromagnetic mechanisms, measuring instruments and relays is the most widely utilized and indispensable soft magnetic alloy in the electrical field. At present, high-grade non-oriented electrical steel strip (sheet) is mainly manufactured using traditional machining methods, such as mechanic shearing and punching. The cold shearing and punching processes produce non-negligible processing stress near the cutting edge, which leads to the deterioration of magnetic characteristics and the increase in motor core loss. Prior research has indicated that this kind of processing stress, as well as its effects on the specific total loss, are dependent on the specific manufacturing equipment and processes used and is essentially unpredictable [1,2]. It is only when the thickness of the punching and shear cutting zone exceeds 60% of the total material thickness that the deterioration degree of the processing stress on the magnetic properties (mainly the specific loss) can be controlled within 5%. At the same time, the punching and shearing dies are needed to be customized and regularly maintained, resulting in a gradual mismatch between the processing flexibility and efficiency and the current industrial development trend. With the deep integration of informatization and industrialization, laser cutting technology has gradually entered the modern processing and manufacturing industry, and has been applied more and more widely and deeply because of its advantages such as high processing efficiency, smooth cutting surface, good processing flexibility and easy integration with numerical control technology.

In recent years, laser cutting technology has been continuously promoted in the electrical steel industry, and a lot research has been carried out to investigate the influencing mechanisms of laser cutting models and parameters on the quality of the cutting edge [3,4,5,6]. The local microstructure, crystallographic texture, chemical composition, inclusion fraction as well as the stress state near the edge can be changed in the melting and solidification process induced by laser cutting [7,8]. These changes demonstrate the complicated effects on the magnetic and mechanical properties. Many studies show that compared with mechanical cutting, the heat affected zone (HAZ) produced by laser cutting deteriorates the specific loss of high-grade non-oriented electrical steel [9,10,11]. However, some researchers argue that laser cutting is superior to mechanical cutting in terms of the magnetic performance of the laminations [12,13]. Hofmann et al. [9] conducted a quantitative analysis of the degradation of the magnetic properties of M330-35A high-grade non-oriented electrical steel sheet (Si ≈ 2.8 wt%) caused by laser, guillotine and spark erosion cutting. It found that the laser cutting (CO_2_ gas laser, cutting power 1500 W, cutting speed 20 m/min and focus diameter 125 μm) increased the losses by more than 100% over a wide working range and was the most harmful cutting technique. By comparing the total specific loss of 0.35 mm high-grade non-oriented electrical steel containing 3% Si manufactured using the five cutting methods, including shearing, punching, laser cutting, wire cutting and water cutting, Shi et al. [10] found that laser cutting showed the highest iron loss due to the residual stress produced by the rapid heating and cooling during the laser cutting process. Saleem et al. [7] gave more attention to the microstructural changes due to laser cutting and its relation to the magnetic properties. They found that laser cutting demonstrated non-linear effects on permeability, with the maximum change in permeability at around 0.6 T–1.0 T. The drop in permeability was attributed to the residual stress and the microstructure modification was induced by laser cutting [8]. Meanwhile, different from the large slab-like domains that exist in the matrix, a very fine striped domain structure perpendicular to the cutting edge was found in the grains close to the edge, which indicates the presence of a stress component perpendicular to the cutting edge [14].

The quantitative study of the influences of laser cutting parameters on the magnetic quality is difficult mainly due to the large number of variables of laser cutting, the varying types of the electrical steel and the limited size of the affected area from the cutting edge—which is too small to examine the microstructure in detail and to precisely measure local magnetic properties [15,16,17]. Most of these studies mentioned above only characterized the change of magnetic properties of non-oriented silicon steel strip (sheet) of the same steel grade (brand) under a certain laser cutting process. In fact, the effects of laser cutting on the microstructure and magnetic properties is closely related to the material properties and the laser cutting parameters. Therefore, in order to solve the complex industrial technical problem of ‘laser cutting may cause the deterioration of magnetic properties of non-oriented electrical steel materials’; to promote its large-scale application in the related field of electrical steel product manufacturing (due to the advantages of laser cutting technology); and to improve the production efficiency and intelligent manufacturing level of the entire electrical steel industry ecosystem and other new areas [18,19,20]; the present work aims to establish the influencing mechanisms of the laser cutting parameters on the magnetic properties of high-grade non-oriented electrical steel based on a serial experiment, and explore the feasibility of laser cutting technology applied to the manufacturing field of high-grade non-oriented electrical steels.

## 2. Experimental Materials and Methods

### 2.1. Materials

The 50W350 high-grade non-oriented electrical steel with 2.91 wt% Si and density of 7.65 g/cm^3^ was chosen for experiments in the present work.

### 2.2. Sample Preparation

One group of samples were sheared using the common flat blade shearing machine with the blade gap of about 2 mm to ensure that the cutting edge was free of burrs. Another group of samples were cut using the YLR-2000 fiber laser. The laser frequency was set as 5000 Hz. The duty-cycle was set to 100%. The nozzle diameter was 1.5 μm. N_2_ was chosen as the auxiliary gas and the kerf width was set to 0.11 mm. Laser cutting height was set to 1 mm. The other cutting parameters were designed based on the Box–Behnken response surface design (as shown in Table 1). A total of 16 samples (8 pieces parallel and 8 pieces normal to the rolling direction) with a size of 320 mm × 30 mm × 0.5 mm were needed for one round of magnetic properties measurement conducted on an Epstein tester. For each compared pair (laser cutting vs. mechanic shearing), the samples were machined at the same corresponding positions along the rolling direction and transverse direction of the steel sheets, respectively. The steel sheet before cutting, the samples after cutting, the mechanic shearing system and the laser cutting system are shown in Figure 1.

### 2.3. Magnetic Properties Measurement and Evaluation

The magnetic properties were measured on the same AC magnetic tester using the Epstein square ring method according to the national standard IEC 60404-2 “Magnetic materials-Part 2: Methods of measurement of the magnetic properties of electrical steel strip and sheet by means of an Epstein frame” [21]. The width of the Epstein square was 25 mm (see Figure 2). The specific total loss (*P_s_*), specifically *P_1.0/50_* and *P_1.5/50_*, and peak magnetic induction intensity (*B_m_*), specifically *B_2500_* and *B_5000_*, could then be examined. Here, *P_1.0/50_* and *P_1.5/50_* represent the specific total loss measured at the frequency of 50 Hz and maximum magnetic induction intensity of 1.0 T and 1.5 T, respectively. *B_2500_* and *B_5000_* represent the magnetic induction intensity peak measured at the magnetic field intensity of 2500 A/m and 5000 A/m, respectively. Two evaluation criteria were designed in this work as expressed in Equations (1) and (2). Here, positive Δ*P_s_* and/or Δ*B_m_* indicates that laser cutting is superior to shearing in regard to the specific magnetic properties. The stress relief annealing (SRA, 750 °C for 2 h under N_2_ atmosphere) was conducted on the manufactured samples to relieve the strain and stress with the aim to recover the magnetic properties:(1)$$\Delta P_{s}=\frac{P_{s}{}_{\left(S\right)}-P_{s}{}_{\left(L\right)}}{P_{s}{}_{\left(S\right)}}\times 100\%$$
(2)$$\Delta B_{m}=\frac{B_{m}{}_{\left(S\right)}-B_{m}{}_{\left(L\right)}}{B_{m}{}_{\left(S\right)}}\times 100\%$$
where *P_s(S)_* and *P_s(L)_* represent specific total loss of shearing and laser cutting, respectively; *B_m(S)_* and *B_m(L)_* represent the peak values of the magnetic polarization of shearing and laser cutting, respectively.

### 2.4. Observation of Cutting Edge and Magnetic Domain

The cutting edge was directly observed using optical metallographic microscope (ZEISS AX10) and scanning electron microscope (SEM, FEI Quanta FEG450). The magnetic domain near the cutting edge was observed by means of Fe_3_O_4_ magnetic fluid. The samples sized 20 mm × 30 mm × 0.5 mm were ground, polished, successively etched with 8% nital (8 mL of HNO_3_ + 92 mL of C_2_H_5_OH) and 5% copper sulfate solution and followed by ethanol solution cleaning. Fe_3_O_4_ magnetic fluid was dripped onto the interested surface of the horizontally-placed samples and, finally after air drying, optical observation was conducted to examine the magnetic domain distribution near the cutting edge.

### 2.5. Optimization of the Laser Cutting Parameters

The response surface model (as shown in Equation (3)) was first utilized to fit the relationship between the experimental Δ*Ps* and the corresponding four laser cutting parameters, i.e., laser power, cutting speed, auxiliary gas N_2_ pressure and defocusing amount; and then to predict the optimal combination of the cutting parameters based on the least square method with *P* less than 0.05:(3)$$\Delta Ps=a_{1}y_{1}^{2}+a_{2}y_{1}y_{2}+a_{3}y_{1}y_{4}+a_{4}y_{1}+b_{1}y_{2}^{2}+b_{2}y_{2}y_{3}+b_{3}y_{2}y_{4}+c_{1}y_{3}^{2}+c_{2}y_{3}y_{4}+c_{3}y_{3}+d_{1}y_{4}+C$$
where *a*_1_, *a*_2_, *a*_3_, *a*_4_, *b*_1_, *b*_2_, *b*_3_, *c*_1_, *c*_2_, *c*_3_ and *d*_1_ are fitted parameters, and *C* is a constant; *y*_1_ is the averaged laser power, *y*_2_ is laser cutting speed, *y*_3_ is laser defocus value and *y*_4_ is auxiliary gas pressure.

## 3. Results and Discussion

### 3.1. Magnetic Properties under Different Laser Cutting Parameters

Figure 3 shows the Δ*Ps* distribution of the specific total loss of the 50W350 high-grade non-oriented electrical steel manufactured in the 26 laser cutting parameter combinations as shown in Table 1. It is found that Δ*P_1.0/50_* is always negative, with the highest difference at −12.6% and the lowest difference at −4.1%, which indicates that mechanic shearing is superior to laser cutting in regard to this evaluation index. The highest and lowest values of Δ*P_1.0/50_* are -12.6% and -4.1%, respectively. However, regarding the evaluation index Δ*P_1.5/50_*, the difference between laser cutting and mechanic shearing narrowed. In some cases, Δ*P_1.5/50_* is positive with the highest value at 0.7%, which indicates that laser cutting is superior to mechanic shearing and that laser cutting could be the better choice for the working environment with a maximum magnetic induction intensity of 1.5 T.

Figure 4 shows the Δ*B_s_* distribution of the magnetic induction intensity peak of the 50W350 high-grade non-oriented electrical steel manufactured in the 26 laser cutting parameter combinations as shown in Table 1. It is found that both Δ*B_2500_* and Δ*B_5000_* are negative and are in almost the same distribution—but with small absolute values for all of the parameter combinations. The highest and lowest differences are −0.56% and −0.29% for index Δ*B_2500_*, respectively, and the highest and lowest differences are −0.30% and −0.02% for index Δ*B_2500_*, respectively, which indicates that there is no significant difference for mechanic shearing and laser cutting regarding these evaluation indices. The influence of individual laser cutting parameters on Δ*Ps* of the specific total loss of 50W350 high-grade non-oriented electrical steel is shown in Figure 5. It is found that in the present work, the auxiliary gas N_2_ pressure exhibits a nearly linear relationship with the Δ*Ps*; and the laser power, cutting speed and defocus amount demonstrate the non-relationships with Δ*Ps;* which indicates that the parameters should be optimized to achieve the lowest specific total loss.

### 3.2. The Effects of Cutting Parameters on the B-H, H-B and μ-B Curves

Generally speaking, the specific total loss of ferromagnetic materials is mainly composed of the hysteresis loss (*P_h_*), eddy current loss (*P_e_*) and residual loss (*P_r_*). The hysteresis loss is proportional to the area of the static hysteresis loop of the investigated material—depending on the energy required for the magnetization. The eddy current loss is related to the characteristics of the material, such as the thickness, the resistivity and so on. The residual loss is generally believed to be caused by the movement of the domain wall, which cannot be accurately quantified at present [22,23]. As shown in Figure 2, the samples manufactured using the laser cutting parameter combinations No. 7 and No. 21 show the lowest and the highest specific total loss. Therefore, in this section, the macro-analysis of cutting edge, the *B-H*, *H-B* and *μ-B* curves of the No. 7 and No. 21 samples are further examined as the representative samples.

Compared with the No. 21 sample, the cutting speed of the No. 7 sample was lowered by 5 m/min, the pressure of the auxiliary gas N_2_ was increased by 0.5 MPa and the defocusing amount was increased by 0.5 mm. Figure 6a,b shows that the cutting edge of the two samples under the laser cutting is relatively flat but with visible molten columns. Sample No. 7 exhibits the relatively finer and denser molten columns, which is believed to be more attributed to the lower cutting speed. Figure 6c,d demonstrates that the plastic deformation near the cut line (upside of the figures) can be clearly seen in the sample manufactured using mechanic shearing. Figure 7 further demonstrates that although there were processing marks on the cut edge induced by the laser cutting process, the quality of the cut edge from laser cutting is much better than those processed by mechanic shearing, even with the observable grain boundaries. In addition, it is found that the quality of the cut edge closely depends on the laser cutting parameters.

Sample No. 7 is manufactured with the laser cutting parameters of cutting speed 20 m/min, laser power 1400 W, auxiliary gas pressure 1.50 MPa and defocus value 0.5 mm. Sample No. 21 is manufactured with laser cutting parameters of cutting speed 25 m/min, laser power 1400 W, auxiliary gas pressure 1.00 Mpa and defocus value 0.0 mm.

Figure 8, Figure 9 and Figure 10 demonstrate that the magnetic properties of the samples manufactured using mechanic shearing are highly stable since all the curves showed limited differences. Figure 8 shows that there is an obvious area increase in the hysteresis loops of the samples manufactured using laser cutting. However, the area increase is relatively narrow as the magnetic induction intensity increased from 1.0 T to 1.5 T as shown in Figure 9. Meanwhile, the No. 21 sample manufactured using laser cutting always showed the deteriorated magnetic properties. To further understand the underlying reasons, the magnetizing curve and permeability curve of the No. 7 and No. 21 samples processed using laser cutting and mechanic shearing are examined as shown in Figure 10. It is clear that when the magnetic induction intensity is lower than about 1.3 T, the magnetic permeabilities of both the No. 7 and No. 21 samples are smaller than those of their counterpart samples processed using mechanic shearing. When the magnetic induction intensity exceeds about 1.3 T, the permeability of No. 7 turns out to be the highest. Meanwhile, when the magnetic induction intensity exceeds about 1.4 T, the permeability of No. 7 surpasses those of the samples processed using mechanic shearing. In addition, the permeability of No. 7 is larger than that of No. 21 within the whole curve. The magnetizing curve also shows that the No. 7 and No. 21 samples are more difficult to be magnetized at the lower magnetic field intensify but easier to be magnetized at the higher magnetic field intensify. These results indicate that sample No. 21 could produce a higher specific total loss than sample No. 7. Meanwhile, the No. 7 and No. 21 samples could improve the index Δ*P_1.5/50_* compared with the index Δ*P_1.0/50_* because their permeability at the higher magnetic induction intensity bypasses the samples manufactured using mechanic shearing, as shown Figure 3, Figure 8 and Figure 9.

It has been shown that the underlying deterioration mechanisms in magnetic properties for mechanic shearing and laser cutting are different. In the case of mechanic shearing, there is a clearly changed microstructure region near the cutting edge due to the elastic and plastic deformation [24]. Different from the mechanic shearing, laser cutting involves the melting and solidification process of the matrix, which could alter the local microstructure, induce residual stress near the cutting edge and then deteriorate local magnetic properties [7,8]. However, a number of works have proved that the laser cutting did not—at least obviously—alter the microstructure morphology as well as the hardness near the cutting edge [10,15,25,26]. By comparing the shearing, laser cutting, wire electric discharge machining, punching and abrasive water jet cutting, it was found that the hardness was not obviously changed along the distance from the cutting edge to the matrix with the depth of 2400 um [10]. Meanwhile, it was also found that at the lower magnetic induction intensity, the sample prepared using laser cutting showed the lowest permeability but bypassed some of the others at the higher magnetic induction intensity, which is consistent with the present work. On the other side, the almost unchanged hardness near the cutting edge indicates that the residual stress near the cutting edge should be limited. Therefore, the magnetic properties loss induced by the laser cutting could not be mainly attributed to the residual stress, which is believed to be very harmful to the magnetic properties [8,14]. Another work experimentally observed that a region with no slab-like domains existed very near the cutting edge, which is referred to as the magnetically hardened zone [9,11], and a region with the very fine striped domain structure perpendicular to the cutting edge was found connecting the magnetically hardened zone and the unaffected zone, which is markedly different from the large slab-like domains existing in the matrix [14]. Therefore, it can be expected that the changed domain structure near the cutting edge induced by the laser cutting process should dominate the magnetic loss examined in the present work.

As discussed above, for mechanic shearing, high residual stress/strain could be preserved near the cutting edge [10]. Therefore, the magnetic properties of the sample manufactured using mechanic shearing could be recovered after the residual stress/strain was relieved by annealing. The stress relief annealing (SRA), which can relieve the strain and stress, is believed to show a positive effect in recovering the magnetic properties of the manufactured samples [27]. Figure 11 shows the hysteresis curves of the No. 7 and No. 21 samples before and after annealing at the magnetic induction intensity of 1.0 T. It is obvious that the magnetic properties of the samples manufactured using laser cutting and mechanic shearing are both significancy improved by annealing at 750 °C for 2 h. Meanwhile, it is found that after annealing the hysteresis curves of No. 7 manufactured using laser cutting and mechanic shearing are almost overlapped with each other, which indicates that the effects of the heat affected zone (HAZ) [11] near the cutting edge formed during the laser cutting process can be omitted in the No. 7 sample. However, for the No. 21 sample, the magnetic properties of the sample manufactured using laser cutting are lower than that of the sample manufactured using mechanic shearing—possibly due to the effects of the heat affected zone (HAZ) [8,11]. Meanwhile, to discover the underlying mechanism in the recovering of magnetic properties of the samples manufactured using laser cutting, the magnetic domain distribution before and after annealing was examined as shown in Figure 12. It is obvious that the magnetic domain is not clear near the cutting edge before the annealing, whereas after the annealing the magnetic domain is obviously observed near the cutting edge. Therefore, it can also be concluded that the thermal stress during the laser cutting changed the magnetic domain structure near the cutting edge, which dominated the magnetic loss of the sample manufactured using laser cutting.

### 3.3. Optimization of the Laser Cutting Parameters

As shown in Figure 3, Figure 4 and Figure 5, there is a non-linear relationship between the laser cutting parameters and the magnetic properties, and as shown in Figure 11, the after-annealing samples manufactured using laser cutting and mechanic shearing could be recovered at almost the same level, which indicates that the magnetic properties of the samples for laser cutting could be significantly improved by optimizing the laser cutting parameters. The set target of relative deviation of specific total loss Δ*P_1.0/50_* is infinitely close to 0.0% with Δ*P_1.5/50_* fluctuation within ±2.0%, and the response surface model is optimized using the model introduced in Section 2.4. A combination of the optimized parameters were obtained with the cutting speed of 20 m/min, the laser power of 1200 W, the auxiliary gas N_2_ pressure of 1.5 MPa and the defocusing amount of 0.14 mm. Three runs of the verification experiment were conducted and the results are shown in Table 2. It shows that the index Δ*P_1.0/50_* has been significantly improved from −4.1%, the best score before the parameters optimization, to 3.7% (−3.4% and −3.5%) with the parameters optimization, with some limited improvement in the index Δ*P_1.5/50_*, which indicates that the magnetic properties of the samples manufactured using the laser cutting can be significantly improved with reasonable parameter optimization.

## 4. Summary


(1)The laser cutting process changed the magnetic domain structures near the cutting edge, which dominated the hysteresis loss increment compared with the annealed samples for the investigated 50W350 high-grade non-oriented electrical steel.(2)Compared to mechanic shearing, the influencing of laser cutting on permeability depend on the magnetic induction intensity with a critical value of about 1.3 T. At the lower magnetic induction intensity (less than 1.3 T) the permeability was significantly decreased, whereas the permeability was slightly enhanced at the higher magnetic induction intensity (larger than 1.3 T), which induced the lower score at the index of Δ*P_1.0/50_* but improved the score at the index Δ*P_1.5/50_*.(3)The parameters of laser cutting demonstrated non-linear effects on the magnetic properties of the cut samples. The mechanic shearing is superior to laser cutting regarding the index *P_1.0/50_* in all of the cases. However, laser cutting is superior to mechanic shearing in some cases regarding index *P_1.5/50_*. Meanwhile, both *P_1.0/50_* and *P_1.5/50_* can be further improved by optimizing the laser cutting parameters.(4)Laser cutting is completely acceptable for 50W350 high-grade non-oriented electrical steel. If the working magnetic induction intensity is more than 1.3 T, the optimized laser cutting parameters can be directly utilized. If the working magnetic induction intensity is less than 1.3 T, laser cutting may result in a 3.5% deterioration compared with mechanic shearing in regard to index *P_1.0/50_*. However, stress relief annealing can be applied to restore the magnetic properties.


## Figures and Tables

**Figure 1 materials-16-01642-f001:**
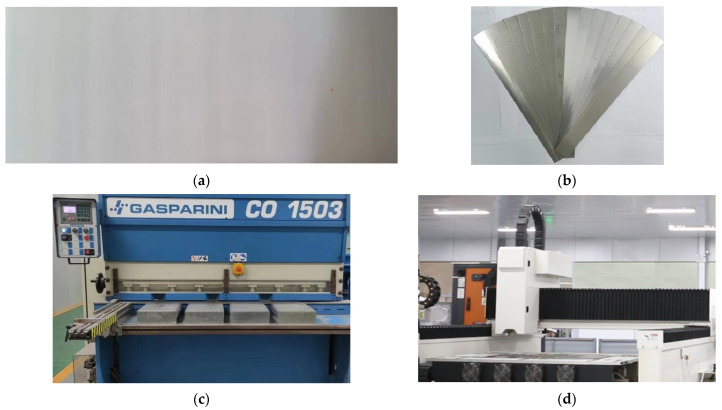
Samples and mechanic shearing/laser cutting system. (**a**) Steel sheet before cutting (400 mm × 1100 mm × 0.5 mm); (**b**) Samples after cutting (320 mm × 30 mm × 0.5 mm); (**c**) Mechanic shearing system; (**d**) Laser cutting system.

**Figure 2 materials-16-01642-f002:**
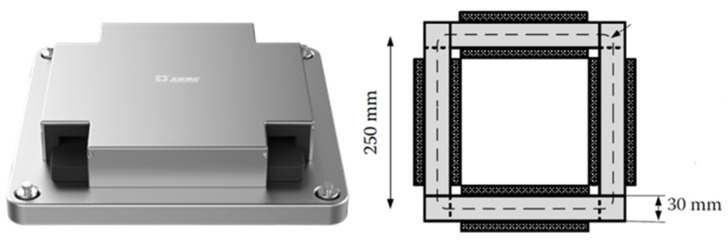
Setup and the schematic diagram of the 25 cm Epstein tester.

**Figure 3 materials-16-01642-f003:**
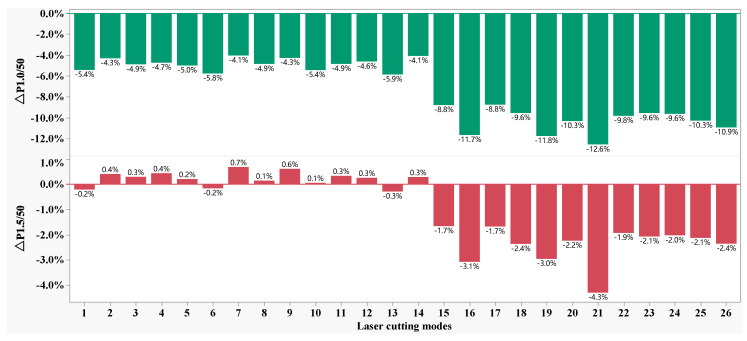
**Δ***P*_s_ distribution under the various laser cutting parameters.

**Figure 4 materials-16-01642-f004:**
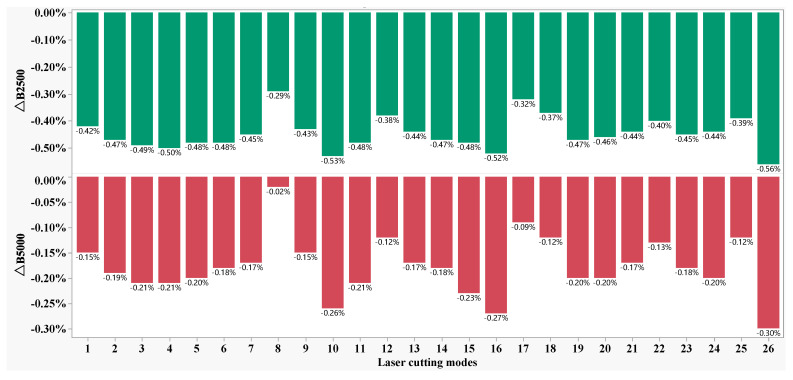
Δ*B*_m_ distribution under the various laser cutting parameters.

**Figure 5 materials-16-01642-f005:**
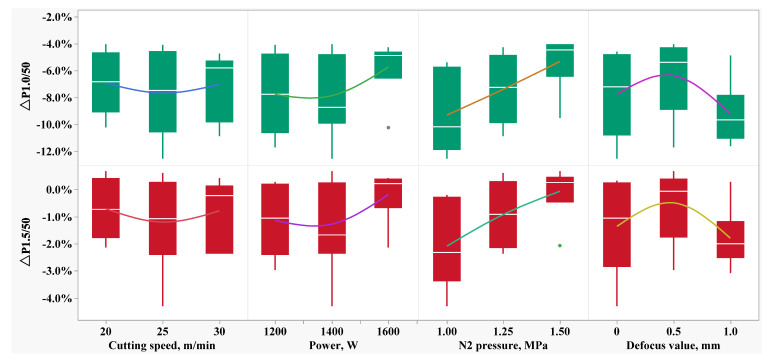
The effects of the individual laser cutting parameters on Δ*P*_s_.

**Figure 6 materials-16-01642-f006:**
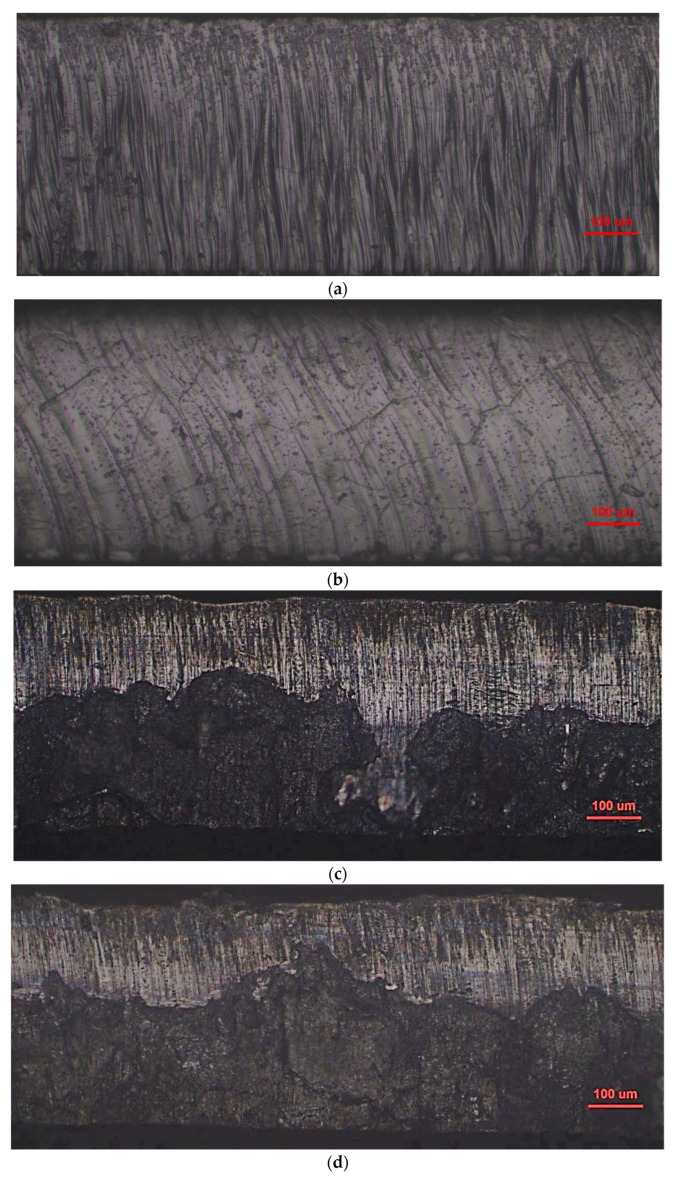
Optical morphology of cutting edge for laser cutting and mechanic shearing. (**a**) No. 7 sample manufactured using laser cutting; (**b**) No. 21 sample manufactured using laser cutting; (**c**) No. 7 sample manufactured using mechanic shearing; (**d**) No. 21 sample manufactured using mechanic shearing.

**Figure 7 materials-16-01642-f007:**
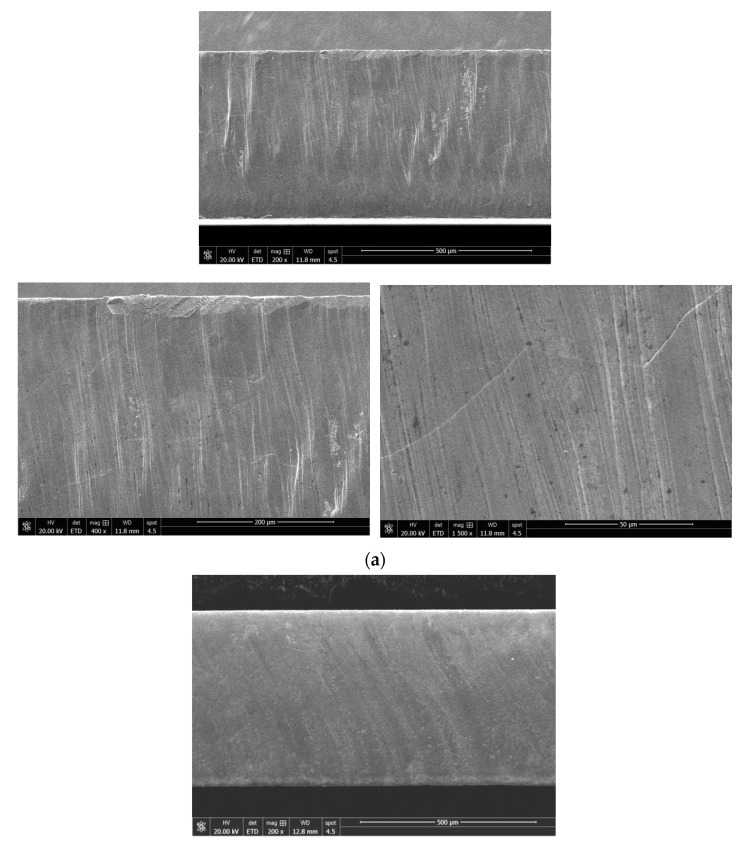
Scanning electron micrographs of cut edges for laser cutting with different magnification times. (**a**) No. 7 sample manufactured using laser cutting; (**b**) No. 21 sample manufactured using laser cutting.

**Figure 8 materials-16-01642-f008:**
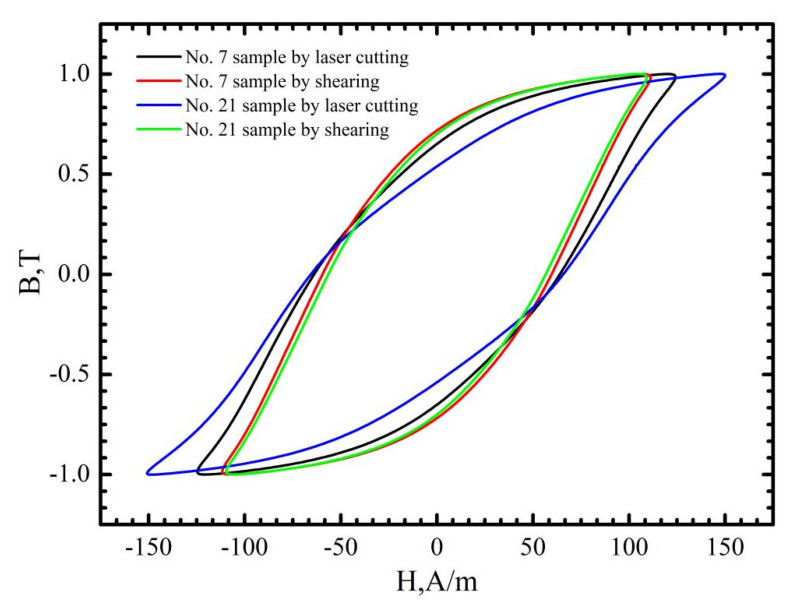
Hysteresis curves at the magnetic induction intensity of 1.0 T of the No. 7 and No. 21 samples processed using laser cutting and mechanic shearing, respectively.

**Figure 9 materials-16-01642-f009:**
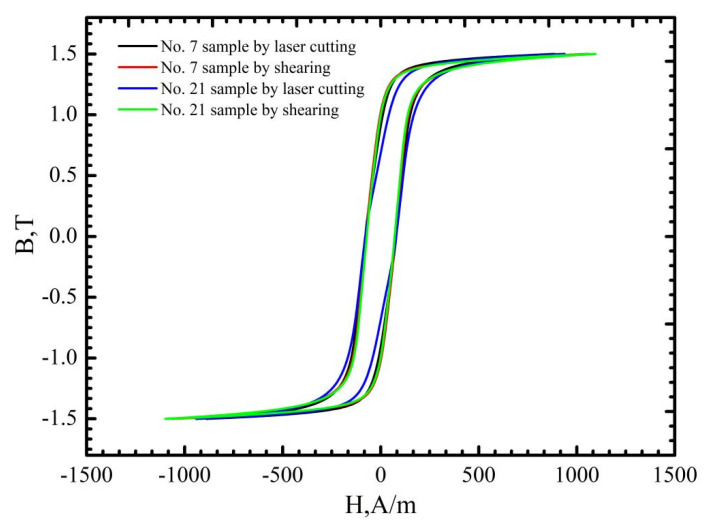
Hysteresis curves at the magnetic induction intensity of 1.5 T of the No. 7 and No. 21 samples processed using laser cutting and mechanic shearing, respectively.

**Figure 10 materials-16-01642-f010:**
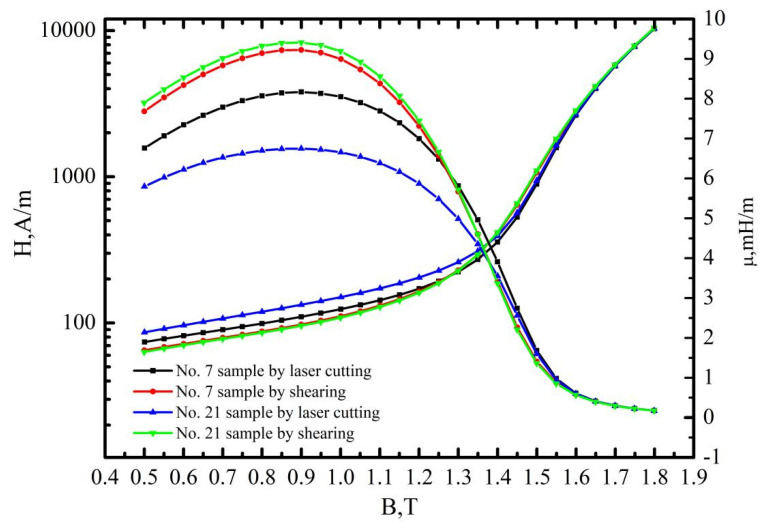
Magnetizing curve and permeability curve of the No. 7 and No. 21 samples processed using laser cutting and mechanic shearing, respectively.

**Figure 11 materials-16-01642-f011:**
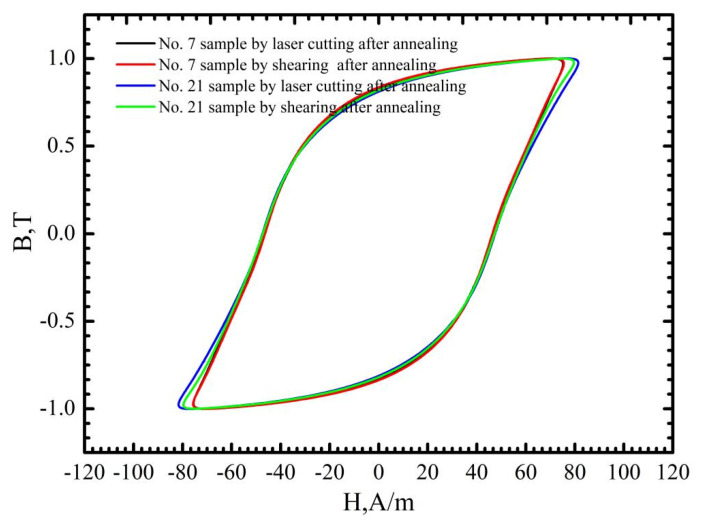
Hysteresis curves of No. 7 and No. 21 samples before and after annealing at the magnetic induction intensity of 1.0 T.

**Figure 12 materials-16-01642-f012:**
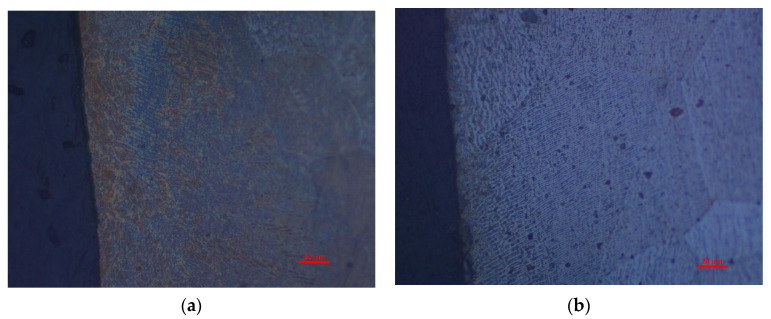
Magnetic domain structures near the cut edge of No. 7 sample (**a**) before and (**b**) after annealing.

**Table 1 materials-16-01642-t001:** Laser cutting parameters utilized in this work. +, 0 and − represent the maximum, middle and minimum values of the varying parameter, respectively.

No.	Laser Model	Cutting Speed, m/min	Power, W	N_2_ Pressure, MPa	Defocus Value, mm
1	0+−0	25	1600	1.00	0.5
2	0++0	25	1600	1.50	0.5
3	0+0+	25	1600	1.25	1.0
4	++00	30	1600	1.25	0.5
5	−−00	20	1200	1.25	0.5
6	+−00	30	1200	1.25	0.5
7	−0+0	20	1400	1.50	0.5
8	0+0−	25	1600	1.25	0.0
9	0	25	1400	1.25	0.5
10	+0+0	30	1400	1.50	0.5
11	−00−	20	1400	1.25	0.0
12	00+−	25	1400	1.50	0.0
13	+0−0	30	1400	1.00	0.5
14	0−+0	25	1200	1.50	0.5
15	−00+	20	1400	1.25	1.0
16	00−+	25	1400	1.00	1.0
17	−0−0	20	1400	1.00	0.5
18	+00−	30	1400	1.25	0.0
19	0−−0	25	1200	1.00	0.5
20	0−0−	25	1200	1.25	0.0
21	00−−	25	1400	1.00	0.0
22	0−0+	25	1200	1.25	1.0
23	00++	25	1400	1.50	1.0
24	0	25	1400	1.25	0.5
25	−+00	20	1600	1.25	0.5
26	+00+	30	1400	1.25	1.0

**Table 2 materials-16-01642-t002:** The evaluation matrix of the three optimized parameter combinations.

No.	*P_1.0/50-s_*	*P_1.0/50-B_*	Δ*P_1.0/50_*	*P_1.5/50-s_*	*P_1.5/50-B_*	Δ*P_1.5/50_*
1	1.186	1.231	−3.7%	2.677	2.666	0.4%
2	1.182	1.223	−3.4%	2.650	2.632	0.7%
3	1.174	1.217	−3.5%	2.648	2.629	0.7%

## Data Availability

The raw/processed data required to reproduce these findings can be shared upon request.

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
