# Peer review of "Effects of Laser Cutting Parameters on the Magnetic Properties of 50W350 High-Grade Non-Oriented Electrical Steel"

_materials, 2023, doi:10.3390/ma16041642_

Round 1
Reviewer 1 Report
In this paper, the results of laser cutting of high-grade non-oriented electrical steel are compared to the same samples cut mechanically. The aim of the presented research is to optimise the laser-cutting process parameters to maintain the magnetic properties of the material. The authors claim that it is possible to some extent.
The research topic is modern given today's electrical motors industry trends. This work is written clearly. The research topic fits the Materials. The manuscript is of good quality, including the quality of the table and most of the figures. The research topic is interesting for scientists, engineers and entrepreneurs.
I have the following comments:
1. In Figures 4 and 5 the parameters of the process for the presented samples should be given. It is difficult to navigate the text to find out all the necessary data.
2. The quality of Fig. 6-9 should be improved by enlarging the text.
Author Response
请参阅附件。

Reviewer 2 Report
1. Please check the format of the manuscript.
2. Introduction section needs to be rewritten with recent literature being added.
3. Please explain the novelty of the present work.
4. Please include the actual picture of the laser cutting setup used.
5. Please mention the constant parameters in this laser cutting.
6. Please include the picture of the samples before and after laser cutting.
7. Please include a few scanning electron micrographs of the samples after laser cutting.
8. The discussion needs to be strengthened. The authors can refer and cite the following works: 10.1088/2051-672X/ac8757, https://doi.org/10.1007/s13369-022-07256-9, https://doi.org/10.1016/j.sna.2008.03.024.
9. The authors can redraw figs. 6 to 8 with some standard plotting softwares.
10. Conclusions to be made more stronger.
Round 2
Reviewer 2 Report
Accept in current form